# Z-Forcing: Training Stochastic Recurrent Networks

**Anirudh Goyal**
MILA, Université de Montréal

**Alessandro Sordoni**
Microsoft Maluuba

**Marc-Alexandre Côté**
Microsoft Maluuba

**Nan Rosemary Ke**
MILA, Polytechnique Montréal

**Yoshua Bengio**
MILA, Université de Montréal

## Abstract

Many efforts have been devoted to training generative latent variable models with autoregressive decoders, such as recurrent neural networks (RNN). Stochastic recurrent models have been successful in capturing the variability observed in natural sequential data such as speech. We unify successful ideas from recently proposed architectures into a stochastic recurrent model: each step in the sequence is associated with a latent variable that is used to condition the recurrent dynamics for future steps. Training is performed with amortised variational inference where the approximate posterior is augmented with a RNN that runs backward through the sequence. In addition to maximizing the variational lower bound, we ease training of the latent variables by adding an auxiliary cost which forces them to reconstruct the state of the backward recurrent network. This provides the latent variables with a task-independent objective that enhances the performance of the overall model. We found this strategy to perform better than alternative approaches such as KL annealing. Although being conceptually simple, our model achieves state-of-the-art results on standard speech benchmarks such as TIMIT and Blizzard and competitive performance on sequential MNIST. Finally, we apply our model to language modeling on the IMDB dataset where the auxiliary cost helps in learning interpretable latent variables.

## 1 Introduction

Due to their ability to capture long-term dependencies, autoregressive models such as recurrent neural networks (RNN) have become generative models of choice for dealing with sequential data. By leveraging weight sharing across timesteps, they can model variable length sequences within a fixed parameter space. RNN dynamics involve a hidden state that is updated at each timestep to summarize all the information seen previously in the sequence. Given the hidden state at the current timestep, the network predicts the desired output, which in many cases corresponds to the next input in the sequence. Due to the deterministic evolution of the hidden state, RNNs capture the entropy in the observed sequences by shaping conditional output distributions for each step, which are usually of simple parametric form, i.e. unimodal or mixtures of unimodal. This may be insufficient for highly structured natural sequences, where there is correlation between output variables at the same step, i.e. *simultaneities* (Boulanger-Lewandowski et al., 2012), and complex dependencies between variables at different timesteps, i.e. *long-term dependencies*. For these reasons, recent efforts recur to highly multi-modal output distribution by augmenting the RNN with stochastic latent variables trained by amortised variational inference, or variational auto-encoding framework (VAE) (Kingma and Welling, 2014; Fraccaro et al., 2016; Kingma and Welling, 2014). The VAE framework allows efficient approximate inference by parametrizing the approximate posterior and generative model with neural networks trainable end-to-end by backpropagation.

Another motivation for including stochastic latent variables in autoregressive models is to infer, from the observed variables in the sequence (e.g. pixels or sound-waves), higher-level abstractions (e.g. objects or speakers). Disentangling in such way the factors of variations is appealing as it would increase high-level control during generation, ease semi-supervised and transfer learning, and enhance interpretability of the trained model (Kingma et al., 2014; Hu et al., 2017).

Stochastic recurrent models proposed in the literature vary in the way they use the stochastic variables to perform output prediction and in how they parametrize the posterior approximation for variational inference. In this paper, we propose a stochastic recurrent generative model that incorporates into a single framework successful techniques from earlier models. We associate a latent variable with each timestep in the generation process. Similar to Fraccaro et al. (2016), we use a (deterministic) RNN that runs *backwards* through the sequence to form our approximate posterior, allowing it to capture the future of the sequence. However, akin to Chung et al. (2015); Bayer and Osendorfer (2014), the latent variables are used to condition the recurrent dynamics for future steps, thus injecting high-level decisions about the upcoming elements of the output sequence. Our architectural choices are motivated by interpreting the latent variables as encoding a "plan" for the future of the sequence. The latent plan is injected into the recurrent dynamics in order to shape the distribution of future hidden states. We show that mixing stochastic forward pass, conditional prior and backward recognition network helps building effective stochastic recurrent models.

The recent surge in generative models suggests that extracting meaningful latent representations is difficult when using a powerful autoregressive decoder, i.e. the latter captures well enough most of the entropy in the data distribution (Bowman et al., 2015; Kingma et al., 2016; Chen et al., 2017; Gulrajani et al., 2017). We show that by using an auxiliary, task-agnostic loss, we ease the training of the latent variables which, in turn, helps achieving higher performance for the tasks at hand. The latent variables in our model are *forced* to contain useful information by predicting the state of the backward encoder, i.e. by predicting the future information in the sequence.

Our work provides the following contributions:

- We unify several successful architectural choices into one generative stochastic model for sequences: backward posterior, conditional prior and latent variables that condition the hidden dynamics of the network. Our model achieves state-of-the-art in speech modeling.

- We propose a simple way of improving model performance by providing the latent variables with an auxiliary, task-agnostic objective. In the explored tasks, the auxiliary cost yielded better performance than other strategies such as KL annealing. Finally, we show that the auxiliary signal helps the model to learn interpretable representations in a language modeling task.

## 2    Background

We operate in the well-known VAE framework (Kingma and Ba, 2014; Burda et al., 2015; Rezende and Mohamed, 2015), a neural network based approach for training generative latent variable models. Let $x$ be an observation of a random variable, taking values in $\mathcal{X}$. We assume that the generation of $x$ involves a latent variable $z$, taking values in $\mathcal{Z}$, by means of a joint density $p_\theta(x, z)$, parametrized by $\theta$. Given a set of observed datapoints $\mathcal{D} = \{x^1, \dots, x^n\}$, the goal of maximum likelihood estimation (MLE) is to estimate the parameters $\theta$ that maximize the marginal log-likelihood $\mathcal{L}(\theta; \mathcal{D})$:

$$\theta^* = \arg\max_\theta \mathcal{L}(\theta; \mathcal{D}) = \sum_{i=1}^{n} \log \int_z p_\theta(x^i, z)\, \mathrm{d}z\,. \tag{1}$$

Optimizing the marginal log-likelihood is usually intractable, due to the integration over the latent variables. A common approach is to maximize a variational lower bound on the marginal log-likelihood. The evidence lower bound (ELBO) is obtained by introducing an approximate posterior $q_\phi(z|x)$ yielding:

$$\log p_\theta(x) \geq \mathop{\mathbb{E}}_{q_\phi(z|x)} \left[ \log \frac{p_\theta(x, z)}{q_\phi(z|x)} \right] = \log p(x) - D_{KL}\big(q_\phi(z|x) \,\|\, p(z|x)\big) = \mathcal{F}(x; \theta, \phi), \tag{2}$$

where KL denotes the Kullback-Leibler divergence. The ELBO is particularly appealing because the bound is tight when the approximate posterior matches the true posterior, i.e. it reduces to the

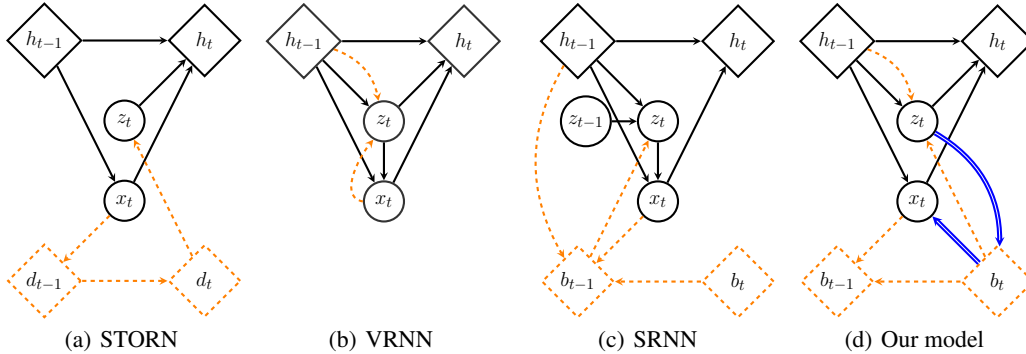

| (a) STORN | (b) VRNN | (c) SRNN | (d) Our model |

Figure 1: Computation graph for generative models of sequences that use latent variables: STORN (Bayer and Osendorfer, 2014), VRNN (Chung et al., 2015), SRNN (Fraccaro et al., 2016) and our model. In this picture, we consider that the task of the generative model consists in predicting the next observation in the sequence, given previous ones. Diamonds represent deterministic states, $z_t$ and $x_t$ are respectively the latent variables and the sequence input at step $t$. Dashed lines represent the computation that is part of the inference model. Double lines indicate auxiliary predictions implied by the proposed auxiliary cost. Differently from VRNN and SRNN, in STORN and our model the latent variable $z_t$ participates to the prediction of the next step $x_{t+1}$.

marginal log-likelihood. The ELBO can also be rewritten as a minimum description length loss function (Honkela and Valpola, 2004):

$$\mathcal{F}(x; \theta, \phi) = \mathbb{E}_{q_\phi(z|x)}\Big[ \log p_\theta(x|z) \Big] - D_{KL}\big( q_\phi(z|x) \,\|\, p_\theta(z) \big), \qquad (3)$$

where the second term measures the degree of dependence between $x$ and $z$, i.e. if $D_{KL}\big( q_\phi(z|x) \,\|\, p_\theta(z) \big)$ is zero then $z$ is independent of $x$. Usually, the parameters of the *generative model* $p_\theta(x|z)$, the *prior* $p_\theta(z)$ and the *inference model* $q_\phi(z|x)$ are computed using neural networks. In this case, the ELBO can be maximized by gradient ascent on a Monte Carlo approximation of the expectation. For particularly simple parametric forms of $q_\phi(z|x)$, e.g. multivariate diagonal Gaussian or, more generally, for *reparamatrizable* distributions (Kingma and Welling, 2014), one can backpropagate through the sampling process $z \sim q_\phi(z|x)$ by applying the *reparametrization* trick, which simulates sampling from $q_\phi(z|x)$ by first sampling from a fixed distribution $u$, $\epsilon \sim u(\epsilon)$, and then by applying deterministic transformation $z = f_\phi(x, \epsilon)$. This makes the approach appealing in comparison to other approximate inference approaches.

In order to have a better generative model overall, many efforts have been put in augmenting the capacity of the approximate posteriors (Rezende and Mohamed, 2015; Kingma et al., 2016; Louizos and Welling, 2017), the prior distribution (Chen et al., 2017; Serban et al., 2017a) and the decoder (Gulrajani et al., 2017; Oord et al., 2016). By having more powerful decoders $p_\theta(x|z)$, one could model more complex distributions over $\mathcal{X}$. This idea has been explored while applying VAEs to sequences $x = (x_1, \ldots, x_T)$, where the decoding distribution $p_\theta(x|z)$ is modeled by an autoregressive model, $p_\theta(x|z) = \prod_t p_\theta(x_t|z, x_{1:t-1})$ (Bayer and Osendorfer, 2014; Chung et al., 2015; Fraccaro et al., 2016). In these models, $z$ typically decomposes as a sequence of latent variables, $z = (z_1, \ldots, z_T)$, yielding $p_\theta(x|z) = \prod_t p_\theta(x_t|z_{1:t-1}, x_{1:t-1})$. We operate in this setting and, in the following section, we present our choices for parametrizing the generative model, the prior and the inference model.

## 3 Proposed Approach

In Figure 1, we report the dependencies in the inference and the generative parts of our model, compared to existing models. From a broad perspective, we use a backward recurrent network for the approximate posterior (akin to SRNN (Fraccaro et al., 2016)), we condition the recurrent state of the forward auto-regressive model with the stochastic variables and use a conditional prior (akin to VRNN (Chung et al., 2015), STORN (Bayer and Osendorfer, 2014)). In order to make better use

of the latent variables, we use auxiliary costs (double arrows) to force the latent variables to encode information about the future. In the following, we describe each of these components.

## 3.1 Generative Model

**Decoder**   Given a sequence of observations $x = (x_1, \ldots, x_T)$, and desired set of labels or predictions $y = (y_1, \ldots, y_T)$, we assume that there exists a corresponding set of stochastic latent variables $z = (z_1, \ldots, z_T)$. In the following, without loss of generality, we suppose that the set of predictions corresponds to a shifted version of the input sequence, i.e. the model tries to predict the next observation given the previous ones, a common setting in language and speech modeling (Fraccaro et al., 2016; Chung et al., 2015). The generative model couples observations and latent variables by using an autoregressive model, i.e. by exploiting a LSTM architecture (Hochreiter and Schmidhuber, 1997), that runs through the sequence:

$$h_t = \overrightarrow{f}(x_t, h_{t-1}, z_t). \tag{4}$$

The parameters of the conditional probability distribution on the next observation $p_\theta(x_{t+1}|x_{1:t}, z_{1:t})$ are computed by a multi-layered feed-forward network that conditions on $h_t$, $f^{(o)}(h_t)$. In the case of continuous-valued observations, $f^{(o)}$ may output the $\mu, \log \sigma$ parameters of a Gaussian distribution, or the categorical proportions in the case of one-hot predictions. Note that, even if $f^{(o)}$ is a simple unimodal distribution, the marginal distribution $p_\theta(x_{t+1}|x_{1:t})$ may be highly multimodal, due to the integration over the sequence of latent variables $z$. Note that $f^{(o)}$ does not condition on $z_t$, i.e. $z_t$ is not directly used in the computation of the output conditional probabilities. We observed better performance by avoiding the latent variables from directly producing the next output.

**Prior**   The parameters of the prior distribution $p_\theta(z_t|x_{1:t}, z_{1:t-1})$ over each latent variable are obtained by using a non-linear transformation of the previous hidden state of the forward network. A common choice in the VAE framework is to use Gaussian latent variables. Therefore, $f^{(p)}$ produces the parameters of a diagonal multivariate Gaussian distribution:

$$p_\theta(z_t|x_{1:t}, z_{1:t-1}) = \mathcal{N}(z_t; \mu_t^{(p)}, \sigma_t^{(p)}) \quad \text{where} \quad [\mu_t^{(p)}, \log \sigma_t^{(p)}] = f^{(p)}(h_{t-1}). \tag{5}$$

This type of conditional prior has proven to be useful in previous work (Chung et al., 2015).

## 3.2 Inference Model

The inference model is responsible for approximating the true posterior over the latent variables $p(z_1, \ldots, z_T|x)$ in order to provide a tractable lower-bound on the log-likelihood. Our posterior approximation uses a LSTM processing the sequence $x$ backwards:

$$b_t = \overleftarrow{f}(x_{t+1}, b_{t+1}). \tag{6}$$

Each state $b_t$ contains information about the future of the sequence and can be used to shape the approximate posterior for the latent $z_t$. As the forward LSTM uses $z_t$ to condition future predictions, the latent variable can directly inform the recurrent dynamics about the future states, acting as a "plan" of the future in the sequence. This information is channeled into the posterior distribution by a feed-forward neural network $f^{(q)}$ taking as input both the previous forward state $h_{t-1}$ and the backward state $b_t$:

$$q_\phi(z_t|x) = \mathcal{N}(z_t; \mu_t^{(q)}, \sigma_t^{(q)}) \quad \text{where} \quad [\mu_t^{(q)}, \log \sigma_t^{(q)}] = f^{(q)}(h_{t-1}, b_t). \tag{7}$$

By injecting stochasticity in the hidden state of the forward recurrent model, the true posterior distribution for a given variable $z_t$ depends on all the variables $z_{t+1:T}$ after $z_t$ through dependence on $h_{t+1:T}$. In order to formulate an efficient posterior approximation, we drop the dependence on $z_{t+1:T}$. This is at the cost of introducing intrinsic bias in the posterior approximation, e.g. we may exclude the true posterior from the space of functions modelled by our function approximator. This is in contrast with SRNN (Fraccaro et al., 2016), in which the posterior distribution factorizes in a tractable manner at the cost of not including the latent variables in the forward autoregressive dynamics, i.e. the latent variables don't condition the hidden state, but only help in shaping a multi-modal distribution for the current prediction.

### 3.3 Auxiliary Cost

In various domains, such as text and images, it has been empirically observed that it is difficult to make use of latent variables when coupled with a strong autoregressive decoder (Bowman et al., 2015; Gulrajani et al., 2017; Chen et al., 2017). The difficulty in learning meaningful latent variables, in many cases of interest, is related to the fact that the abstractions underlying observed data may be encoded with a smaller number of bits than the observed variables. For example, there are multiple ways of picturing a particular "cat" (e.g. different poses, colors or lightning) without varying the more abstract properties of the concept "cat". In these cases, the maximum-likelihood training objective may not be sensitive to how well abstractions are encoded, causing the latent variables to "shut off", i.e. the local correlations at the pixel level may be too strong and bias the learning process towards finding parameter solutions for which the latent variables are unused. In these cases, the posterior approximation tends to provide a too weak or noisy signal, due to the variance induced by the stochastic gradient approximation. As a result, the decoder may learn to ignore $z$ and instead to rely solely on the autoregressive properties of $x$, causing $x$ and $z$ to be independent, i.e. the KL term in Eq. 2 vanishes.

Recent solutions to this problem generally propose to reduce the capacity of the autoregressive decoder (Bowman et al., 2015; Bachman, 2016; Chen et al., 2017; Semeniuta et al., 2017). The constraints on the decoder capacity inherently bias the learning towards finding parameter solutions for which $z$ and $x$ are dependent. One of the shortcomings with this approach is that, in general, it may be hard to achieve the desired solutions by architecture search. Instead, we investigate whether it is useful to keep the expressiveness of the autoregressive decoder but *force* the latent variables to encode useful information by adding an auxiliary training signal for the latent variables alone. In practice, our results show that this auxiliary cost, albeit simple, helps achieving better performance on the objective of interest.

Specifically, we consider training an additional conditional generative model of the backward states $b = \{b_1, \ldots, b_T\}$ given the forward states $p_\xi(b|h) = \int_z p_\xi(b, z|h) dz \geq \mathbb{E}_{q_\xi(z|b,h)}[\log p_\xi(b|z) + \log p_\xi(z|h) - \log q_\xi(z|b,h)]$. This additional model is also trained through amortized variational inference. However, we share its prior $p_\xi(z|h)$ and approximate posterior $q_\xi(z|b, h)$ with those of the "primary" model ($b$ is a deterministic function of $x$ per Eq. 6 and the approximate posterior is conditioned on $b$). In practice, we solely learn additional parameters $\xi$ for the decoding model $p_\xi(b|z) = \prod_t p_\xi(b_t|z_t)$. The auxiliary reconstruction model trains $z_t$ to contain relevant information about the future of the sequence contained in the hidden state of the backward network $b_t$:

$$p_\xi(b_t|z_t) = \mathcal{N}(\mu_t^{(a)}, \sigma_t^{(a)}) \quad \text{where} \quad [\mu_t^{(a)}, \log \sigma_t^{(a)}] = f^{(a)}(z_t), \tag{8}$$

By means of the auxiliary reconstruction cost, the approximate posterior and prior of the primary model is trained with an additional signal that may help with escaping local minima due to short term reconstructions appearing in the lower bound, similarly to what has been recently noted in Karl et al. (2016).

### 3.4 Learning

The training objective is a regularized version of the lower-bound on the data log-likelihood based on the variational free-energy, where the regularization is imposed by the auxiliary cost:

$$\mathcal{L}(x; \theta, \phi, \xi) = \sum_t \mathop{\mathbb{E}}_{q_\phi(z_t|x)} \Big[ \log p_\theta(x_{t+1}|x_{1:t}, z_{1:t}) + \alpha \log p_\xi(b_t|z_t) \Big]$$
$$- D_{KL}\big(q_\phi(z_t|x_{1:T}) \,\|\, p_\theta(z_t|x_{1:t}, z_{1:t-1})\big). \tag{9}$$

We learn the parameters of our model by backpropagation through time (Rumelhart et al., 1988) and we approximate the expectation with one sample from the posterior $q_\phi(z|x)$ by using reparametrization. When optimizing Eq. 9, we disconnect the gradients of the auxiliary prediction from affecting the backward network, i.e. we don't use the gradients $\nabla_\phi \log p_\xi(b_t|z_t)$ to train the parameters $\phi$ of the approximate posterior: intuitively, the backward network should be agnostic about the auxiliary task assigned to the latent variables. It also performed better empirically. As the approximate posterior is trained only with the gradient flowing through the ELBO, the backward states $b$ may be receiving a weak training signal early in training, which may hamper the usefulness of the auxiliary generative cost, i.e. all the backward states may be concentrated around the zero vector. Therefore,

we additionally train the backward network to predict the output variables in reverse (see Figure 1):

$$\mathcal{L}(x; \theta, \phi, \xi) = \sum_t \mathop{\mathbb{E}}_{q_\phi(z_t|x)} \Big[ \log p_\theta(x_{t+1}|x_{1:t}, z_{1:t}) + \alpha \log p_\xi(b_t|z_t) \Big] + \beta \log p_\xi(x_t|b_t)$$
$$- D_{KL}\big(q_\phi(z_t|x_{1:T}) \, \| \, p_\theta(z_t|x_{1:t}, z_{1:t-1})\big). \tag{10}$$

### 3.5 Connection to previous models

Our model is similar to several previous stochastic recurrent models: similarly to STORN (Bayer and Osendorfer, 2014) and VRNN (Chung et al., 2015) the latent variables are provided as input to the autoregressive decoder. Differently from STORN, we use the conditional prior parametrization proposed in Chung et al. (2015). However, the generation process in the VRNN differs from our approach. In VRNN, $z_t$ are directly used, along with $h_{t-1}$, to produce the next output $x_t$. We found that the model performed better if we relieved the latent variables from producing the next output. VRNN has a "myopic" posterior in such that the latent variables are not informed about the whole future in the sequence. SRNN (Fraccaro et al., 2016) addresses the issue by running a posterior backward in the sequence and thus providing future context for the current prediction. However, the autoregressive decoder is not informed about the future of the sequence through the latent variables. Several efforts have been made in order to bias the learning process towards parameter solutions for which the latent variables are used (Bowman et al., 2015; Karl et al., 2016; Kingma et al., 2016; Chen et al., 2017; Zhao et al., 2017). Bowman et al. (2015) tackle the problem in a language modeling setting by dropping words from the input at random in order to weaken the autoregressive decoder and by annealing the KL divergence term during training. We achieve similar latent interpolations by using our auxiliary cost. Similarly, Chen et al. (2017) propose to restrict the receptive field of the pixel-level decoder for image generation tasks. Kingma et al. (2016) propose to reserve some *free bits* of KL divergence. In parallel to our work, the idea of using a task-agnostic loss for the latent variables alone has also been considered in (Zhao et al., 2017). The authors force the latent variables to predict a bag-of-words representation of a dialog utterance. Instead, we work in a sequential setting, in which we have a latent variable for each timestep in the sequence.

## 4  Experiments

In this section, we evaluate our proposed model on diverse modeling tasks (speech, images and text). We show that our model can achieve state-of-the-art results on two speech modeling datasets: Blizzard (King and Karaiskos, 2013) and TIMIT raw audio datasets (also used in Chung et al. (2015)). Our approach also gives competitive results on sequential generation on MNIST (Salakhutdinov and Murray, 2008). For text, we show that the the auxiliary cost helps the latent variables to capture information about latent structure of language (e.g. sequence length, sentiment). In all experiments, we used the ADAM optimizer (Kingma and Ba, 2014).

### 4.1  Speech Modeling and Sequential MNIST

**Blizzard and TIMIT**  We test our model in two speech modeling datasets. Blizzard consists in 300 hours of English, spoken by a single female speaker. TIMIT has been widely used in speech recognition and consists in 6300 English sentences read by 630 speakers. We train the model directly on raw sequences represented as a sequence of 200 real-valued amplitudes normalized using the global mean and standard deviation of the training set. We adopt the same train, validation and test split as in Chung et al. (2015). For Blizzard, we report the average log-likelihood for half-second sequences (Fraccaro et al., 2016), while for TIMIT we report the average log-likelihood for the sequences in the test set.

In this setting, our models use a fully factorized multivariate Gaussian distribution as the output distribution for each timestep. In order to keep our model comparable with the state-of-the-art, we keep the number of parameters comparable to those of SRNN (Fraccaro et al., 2016). Our forward/backward networks are LSTMs with 2048 recurrent units for Blizzard and 1024 recurrent units for TIMIT. The dimensionality of the Gaussian latent variables is 256. The prior $f^{(p)}$, inference $f^{(q)}$ and auxiliary networks $f^{(a)}$ have a single hidden layer, with 1024 units for Blizzard and 512 units for TIMIT, and use leaky rectified nonlinearities with leakiness $\frac{1}{3}$ and clipped at $\pm 3$ (Fraccaro et al., 2016). For Blizzard, we use a learning rate of 0.0003 and batch size of 128, for TIMIT they are

| Model | Blizzard | TIMIT | Models | MNIST |
|---|---|---|---|---|
| RNN-Gauss | 3539 | -1900 | DBN 2hl (Germain et al., 2015) | $\approx 84.55$ |
| RNN-GMM | 7413 | 26643 | NADE (Uria et al., 2016) | 88.33 |
| VRNN-I-Gauss | $\geq 8933$ | $\geq 28340$ | EoNADE-5 2hl (Raiko et al., 2014) | 84.68 |
| VRNN-Gauss | $\geq 9223$ | $\geq 28805$ | DLGM 8 (Salimans et al., 2014) | $\approx 85.51$ |
| VRNN-GMM | $\geq 9392$ | $\geq 28982$ | DARN 1hl (Gregor et al., 2015) | $\approx 84.13$ |
| SRNN (smooth+res$_q$) | $\geq 11991$ | $\geq 60550$ | DRAW (Gregor et al., 2015) | $\leq 80.97$ |
| | | | PixelVAE (Gulrajani et al., 2016) | $\approx$ **79.02**$^{\blacktriangledown}$ |
| Ours | $\geq 14435$ | $\geq 68132$ | P-Forcing$_{(3\text{-layer})}$ (Goyal et al., 2016) | 79.58$^{\blacktriangledown}$ |
| Ours + `kla` | $\geq 14226$ | $\geq 68903$ | PixelRNN$_{(1\text{-layer})}$ (Oord et al., 2016) | 80.75 |
| | | | PixelRNN$_{(7\text{-layer})}$ (Oord et al., 2016) | 79.20$^{\blacktriangledown}$ |
| Ours + `aux` | $\geq$ **15430** | $\geq 69530$ | MatNets (Bachman, 2016) | 78.50$^{\blacktriangledown}$ |
| Ours + `kla, aux` | $\geq 15024$ | $\geq$ **70469** | | |
| | | | Ours$_{(1\text{ layer})}$ | $\leq 80.60$ |
| | | | Ours + `aux`$_{(1\text{ layer})}$ | $\leq$ **80.09** |

Table 1: On the left, we report the average log-likelihood per sequence on the test sets for Blizzard and TIMIT datasets. "`kla`" and "`aux`" denote respectively KL annealing and the use of the proposed auxiliary costs. On the right, we report the test set negative log-likelihood for sequential MNIST, where $^{\blacktriangledown}$ denotes lower performance of our model with respect to the baselines. For MNIST, we observed that KL annealing hurts overall performance.

0.001 and 32 respectively. Previous work reliably anneal the KL term in the ELBO via a temperature weight during training (KL annealing) (Fraccaro et al., 2016; Chung et al., 2015). We report the results obtained by our model by training both with KL annealing and without. When KL annealing is used, the temperature was linearly annealed from 0.2 to 1 after each update with increments of 0.00005 (Fraccaro et al., 2016).

We show our results in Table 1 (left), along with results that were obtained by models of comparable size to SRNN. Similar to (Fraccaro et al., 2016; Chung et al., 2015), we report the conservative evidence lower bound on the log-likelihood. In Blizzard, the KL annealing strategy (Ours + `kla`) is effective in the first training iterations, but eventually converges to a slightly lower log-likelihood than the model trained without KL annealing (Ours). We explored different annealing strategies but we didn't observe any improvements in performance. Models trained with the proposed auxiliary cost outperform models trained with KL annealing strategy in both datasets. In TIMIT, it appears that there is a slightly synergistic effect between KL annealing and auxiliary cost. Even if not explicitly reported in the table, similar performance gains were observed on the training sets.

**Sequential MNIST** The task consists in pixel-by-pixel generation of binarized MNIST digits. We use the standard binarized MNIST dataset used in Larochelle and Murray (2011). Both forward and backward networks are LSTMs with one layer of 1024 hidden units. We use a learning rate of 0.001 and batch size of 32. We report the results in Table 1 (right). In this setting, we observed that KL annealing hurt performance of the model. Although being architecturally flat, our model is competitive with respect to strong baselines, e.g. DRAW (Gregor et al., 2015), and is outperformed by deeper version of autoregressive models with latent variables, i.e. PixelVAE (gated) (Gulrajani et al., 2016), and deep autoregressive models such as PixelRNN (Oord et al., 2016) and MatNets (Bachman, 2016).

## 4.2 Language modeling

A well-known result in language modeling tasks is that the generative model tends to fit the observed data without storing information in the latent variables, i.e. the KL divergence term in the ELBO becomes zero (Bowman et al., 2015; Zhao et al., 2017; Serban et al., 2017b). We test our proposed stochastic recurrent model trained with the auxiliary cost on a medium-sized IMDB text corpus containing 350K movie reviews (Diao et al., 2014). Following the setting described in Hu et al. (2017), we keep only sentences with less than 16 words and fixed the vocabulary size to 16K words. We split the dataset into train/valid/test sets following these ratios respectively: 85%, 5%, 10%. Special delimiter tokens were added at the beginning and end of each sentence but we only learned to

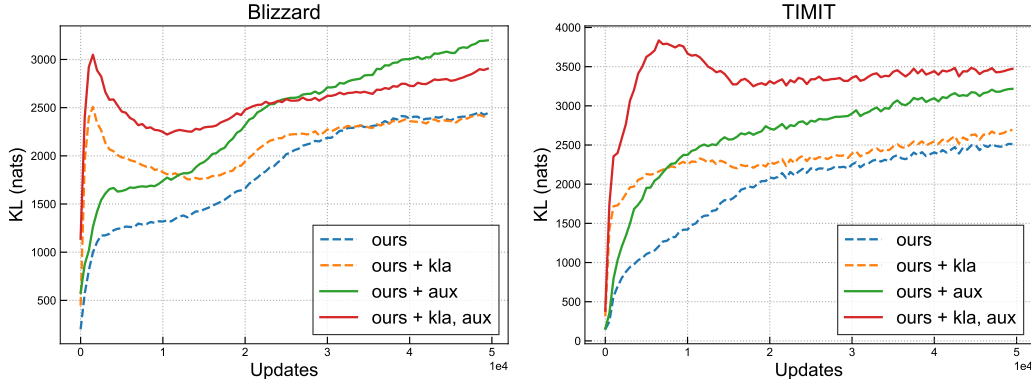

Figure 2: Evolution of the KL divergence term (measured in nats) in the ELBO with and without auxiliary cost during training for Blizzard (left) and TIMIT (right). We plot curves for models that performed best after hyper-parameter (KL annealing and auxiliary cost weights) selection on the validation set. The auxiliary cost puts pressure on the latent variables resulting in higher KL divergence. Models trained with the auxiliary cost (Ours + `aux`) exhibit a more stable evolution of the KL divergence. Models trained with auxiliary cost alone achieve better performance than using KL annealing alone (Ours + `kla`) and similar, or better performance for Blizzard, compared to both using KL annealing and auxiliary cost (Ours + `kla`, `aux`).

| Model | $\alpha, \beta$ | KL | Valid | | Test | |
|---|---|---|---|---|---|---|
| | | | ELBO | IWAE | ELBO | IWAE |
| Ours | 0 | 0.12 | 53.93 | 52.40 | 54.67 | 53.11 |
| Ours + `aux` | 0.0025 | 3.03 | 55.71 | 52.54 | 56.57 | 53.37 |
| Ours + `aux` | 0.005 | 9.82 | 65.03 | 58.13 | 65.84 | 58.83 |

Table 2: IMDB language modeling results for models trained by maximizing the standard evidence lower-bound. We report word perplexity as evaluated by both the ELBO and the IWAE bound and KL divergence between approximate posterior and prior distribution, for different values of auxiliary cost hyperparameters $\alpha, \beta$. The gap in perplexity between the ELBO and IWAE (evaluated with 25 samples) increases with greater KL divergence values.

generate the end of sentence token. We use a single layered LSTM with 500 hidden recurrent units, fix the dimensionality of word embeddings to 300 and use 64 dimensional latent variables. All the $f^{(\cdot)}$ networks are single-layered with 500 hidden units and leaky relu activations. We used a learning rate of 0.001 and a batch size of 32.

Results are shown in Table 2. As expected, it is hard to obtain better perplexity than a baseline model when latent variables are used in language models. We found that using the IWAE (Importance Weighted Autoencoder) (Burda et al., 2015) bound gave great improvements in perplexity. This observation highlights the fact that, in the text domain, the ELBO may be severely underestimating the likelihood of the model: the approximate posterior may loosely match the true posterior and the IWAE bound can correct for this mismatch by tightening the posterior approximation, i.e. the IWAE bound can be interpreted as the standard VAE lower bound with an implicit posterior distribution (Bachman and Precup, 2015). On the basis of this observation, we attempted training our models with the IWAE bound, but observed no noticeable improvement on validation perplexity.

We analyze whether the latent variables capture characteristics of language by interpolating in the latent space (Bowman et al., 2015). Given a sentence, we first infer the latent variables at each step by running the approximate posterior and then concatenate them in order to form a contiguous latent encoding for the input sentence. Then, we perform linear interpolation in the latent space between the latent encodings of two sentences. At each step of the interpolation, the latent encoding is run through the decoder network to generate a sentence. We show the results in Table 3.

| | **this movie is so terrible . never watch ever** | |
|---|---|---|
| *a* | *Argmax* | *Sampling* |
| 0.0 | it 's a movie that does n't work ! | this film is more of a " classic " |
| 0.1 | it 's a movie that does n't work ! | i give it a 5 out of 10 |
| 0.2 | it 's a movie that does n't work ! | i felt that the movie did n't have any |
| 0.3 | it 's a very powerful piece of film ! | i do n't know what the film was about |
| 0.4 | it 's a very powerful story about it ! | the acting is good and the acting is very good |
| 0.5 | it 's a very powerful story about a movie about life | the acting is great and the acting is good too |
| 0.6 | it 's a very dark part of the film , eh ? | i give it a 7 out of 10 , kids |
| 0.7 | it 's a very dark movie with a great ending ! ! | the acting is pretty good and the story is great |
| 0.8 | it 's a very dark movie with a great message here ! | the best thing i 've seen before is in the film |
| 0.9 | it 's a very dark one , but a great one ! | funny movie , with some great performances |
| 1.0 | it 's a very dark movie , but a great one ! | but the acting is good and the story is really interesting |

**this movie is great . i want to watch it again !**

| | **(1 / 10) violence : yes .** | |
|---|---|---|
| *a* | *Argmax* | *Sampling* |
| 0.0 | greetings again from the darkness . | greetings again from the darkness . |
| 0.1 | " oh , and no . | " let 's screenplay it . |
| 0.2 | " oh , and it is . | rating : **** out of 5 . |
| 0.3 | well ... i do n't know . | i do n't know what the film was about |
| 0.4 | so far , it 's watchable . | ( pg-13 ) violence , no . |
| 0.5 | so many of the characters are likable . | just give this movie a chance . |
| 0.6 | so many of the characters were likable . | so far , but not for children |
| 0.7 | so many of the characters have been there . | so many actors were excellent as well . |
| 0.8 | so many of them have fun with it . | there are a lot of things to describe . |
| 0.9 | so many of the characters go to the house ! | so where 's the title about the movie ? |
| 1.0 | so many of the characters go to the house ! | as much though it 's going to be funny ! |

**there was a lot of fun in this movie !**

Table 3: Results of linear interpolation in the latent space. The left column reports greedy argmax decoding obtained by selecting, at each step of the decoding, the word with maximum probability under the model distribution, while the right column reports random samples from the model. $a$ is the interpolation parameter. In general, latent variables seem to capture the length of the sentences.

# 5   Conclusion

In this paper, we proposed a recurrent stochastic generative model that builds upon recent architectures that use latent variables to condition the recurrent dynamics of the network. We augmented the inference network with a recurrent network that runs backward through the input sequence and added a new auxiliary cost that forces the latent variables to reconstruct the state of that backward network, thus explicitly encoding a summary of future observations. The model achieves state-of-the-art results on standard speech benchmarks such as TIMIT and Blizzard. The proposed auxiliary cost, albeit simple, appears to promote the use of latent variables more effectively compared to other similar strategies such as KL annealing. In future work, it would be interesting to use a multitask learning setting, e.g. sentiment analysis as in (Hu et al., 2017). Also, it would be interesting to incorporate the proposed approach with more powerful autogressive models, e.g. PixelRNN/PixelCNN (Oord et al., 2016).

# Acknowledgments

The authors would like to thank Phil Bachman, Alex Lamb and Adam Trischler for the useful discussions. AG and YB would also like to thank NSERC, CIFAR, Google, Samsung, IBM and Canada Research Chairs for funding, and Compute Canada and NVIDIA for computing resources. The authors would also like to express debt of gratitude towards those who contributed to Theano over the years (as it is no longer maintained), making it such a great tool.

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
