[Reviews · NeurIPS 2017]

Reviewer 1



The paper introduces a training technique that encourages autoregressive models based on RNNs to utilize latent variables. More specifically, the training loss of a recurrent VAE with a bidirectional inference network is augmented with an auxiliary loss. The paper is well structured and written, and it has an adequate review of previous work. The technique introduced is heuristic and justified intuitively. The heuristic is backed by empirical results that are good but not ground-breaking. The results for speech modelling are very good, and for sequential MNIST good. The auxiliary loss brings no quantitative improvement to text modelling , but an interesting use for the latent variables is shown in section 5.4. Some concerns: Line 157 seems to introduce a second auxiliary loss that is not shown in L_{aux} (line 153). Is this another term in L_{aux}? Was the backward RNN pretrained on this objective? More details should be included about this. Otherwise, this paper should be rejected as there is plenty of space left to include the details. Line 186: How was the \beta term introduced? What scheduled? Did detail this prove essential? No answers to these questions are provided Line 191: What type of features were used? Raw signal? Spectral? Mel-spectral? Cepstral? Please specify. Line 209: Why the change from LSTM to GRU? Was this necessary for obtaining good results? Were experiments run with LSTM? Line 163: Gradient disconnection here and on paragraph starting on line 127. Can they be shown graphically on Figure 1? Maybe an x on the relevant lines? Figure 2 has very low quality on a print out. Should be improved. Line 212: "architecturaly flat" do the refer to number of layers? Do any of the baseline models include information about the 2D structure of the MNIST examples? specify if any does as this would be advantageous for those models. The section that refers to IWAE: Were the models trained with IWAE or just evaluated? I think it was just evaluated, but it should be more clearly specified. Would it be possible to train using IWAE?

Reviewer 2



The authors of the paper propose a method to improve the training of stochastic recurrent models. More explicitly, they let the approximate posterior for the latent state at time step t depend on the observations of the future–which the authors claim to be novel, but which has been done before. Further, a heuristic in the form of an auxiliary regulariser is presented. ## Major - The idea to let the approximate posterior be based on future time steps goes back to at least [1] who used a bidirectional recognition network - Mostly aesthetically point: The authors wish to encourage the hidden state to carry information about the future; they identify the expressiveness of the likelihood as the problem. One way out would be to then chose a graphical model that reflects this; i.e. through the common Markov assumptions. This is what the authors of [2, 3] do. The paper does not mention this approach. - I feel unsatisfied with proposed auxiliary cost. Some questions are - Do we have a probabilistic interpretation of it? What does adding such a model mean? Maybe it is a proxy to a sequential prior? - Adding the auxiliary costs leads to improved generalisation. But does it also reduce training loss? What about training perplexity? ## References [1] Bayer, Justin, and Christian Osendorfer. "Learning stochastic recurrent networks." arXiv preprint arXiv:1411.7610 (2014). [2] Krishnan, Rahul G., Uri Shalit, and David Sontag. "Deep kalman filters." arXiv preprint arXiv:1511.05121 (2015). [3] Karl, Maximilian, et al. "Deep variational Bayes filters: Unsupervised learning of state space models from raw data." arXiv preprint arXiv:1605.06432 (2016).

Reviewer 3



This paper presents a generative model for sequences and an associated training algorithm based on amortized variational inference. The generative model is equivalent to a nonlinear state-space model and the inference model is based on a RNN running backwards along the sequence. In addition to the variational inference cost function, the authors propose the use of two extra penalty terms in the cost function to encourage the latent variables to reconstruct the state of the RNN of the inference model. The description of the generative model, inference model and auxiliary costs is clear. Comprehensive experiments are performed on speech, image and text sequences. The authors report good performance of their model against many recent models. Since the value of this article is based on this empirical performance, I consider it essential that the authors release the code as they promise in Section 5. The article is well written and tackles the very relevant problem of learning generative models of sequences.